# How Being a Researcher Impacted My Life

**Alix Herer ¹ and Ariel E. Schwartz ²,***

1 Sargent College of Health & Rehabilitation Sciences, Boston University, Boston, MA 02215, USA; alix.herer14@gmail.com
2 Department of Occupational Therapy, Massachusetts General Hospital Institute of Health Professions, Boston, MA 02129, USA
* Correspondence: aschwartz1@mghihp.edu

**Abstract:** Doing research can help people to learn about a lot of different topics. For example, researchers can learn how to work in a group, collect and analyze data, how to make accessible materials, and get to know their own strengths. In this paper, a researcher shares what she learned while working on two research projects about young adults with intellectual/developmental disabilities and co-occurring mental health conditions. The first project was peer mentoring. In this project, we focused on individual needs for someone who may be struggling with mental health. The second project was about workplace disclosure. In this project, we tried to find out if young adults disclose their mental health conditions at work and in job interviews. We explain how we did the projects, how the researcher learned to do research, and what made it easy to learn. We also share about the impact of doing research on the researcher's personal life. Finally, we share why doing inclusive research is important and how to help researchers with disabilities feel like they are valued members of the research team.

**Keywords:** inclusive research; participatory research; intellectual disability; developmental disability; mental health





## 1. Introduction

This paper is about the experience of a co-researcher with a disability doing research. She worked on research with Ariel since 2018. Alix shares her experience on two different projects and why doing research is important to her. We wrote the paper together. Ariel wrote questions for Alix to answer on her own. Then, they met together to talk about what Alix wrote, make edits, and add more details. Usually when they met, Ariel typed everything that Alix said. Then, Ariel looked at everything Alix wrote, organized it, and got Alix's feedback to make sure she did not change the meaning.

We decided that it was not important to write about how to do inclusive research or why inclusive research helps make research better from the perspective of Ariel. Many researchers have already written about this (e.g., Frankena et al. 2019; Schwartz et al. 2020a; Stack and McDonald 2014; Walmsley et al. 2018), especially from the perspective of academics (Strnadová and Walmsley 2018). Alix felt it was very important to write this paper because often papers are written by people without disabilities. This paper is special because it is written by someone with real-life experience explaining how she did the research, which has been done by few other people. While some teams have written together (for a review on this topic, see Strnadová and Walmsley 2018), we only identified one article where an individual co-researcher took the lead in the full manuscript (White and Morgan 2012). Other times, teams of co-researchers wrote about their research process and their experiences (e.g., Abell et al. 2007; Cook et al. 2021; Williams et al. 2005). However, these examples are limited, due to the many challenges of publishing inclusively in academic journals (Riches et al. 2020). In this paper, Alix shares her experiences, what helped her to conduct research, how research impacted her life, why *she* thinks it was important that she was involved, and how she felt as an important member of the research team.

## 2. Study 1: Peer Mentoring

### 2.1. What the Study Was about and What I Did

The peer mentoring study was about a program for individuals with intellectual/developmental disabilities and mental health challenges. In peer mentoring, mentees met with mentors who taught coping strategies and the skills they can utilize in their day-to-day life. We individualized each mentee's plans and activities, so they all reached their own goals.

First, we had to make peer mentoring (Schwartz et al. 2020b). Second, we were peer mentors. Third, we looked at data to find out if the mentoring helped our mentees (Schwartz and Levin 2021).

### 2.1.1. Making the Peer Mentoring Program

Before mentoring happened, we had to learn how to mentor someone like us. We met at Boston University once a week. I worked with three other researchers with disabilities. They all went to the same transition program that I went to. That was cool, because we all knew each other, and I did not have to learn about new people. It was also cool that we were able to work with Boston University graduate students who were around my same age.

Ariel made the activities and the mentoring script. I helped to give feedback on what to keep and what not to keep. We worked together by looking at PowerPoints. We said what pictures and words should or should not be in the peer mentoring materials. We tried the activities before we mentored and gave feedback. I joined afterward, but I know we also did focus groups with young adults all around Massachusetts to see if they liked the peer mentoring activities. Before peer mentoring started, we also made videos about mental health to show the mentees. That was really fun!

When we were making mentoring, it was helpful to have interactive activities, like the PowerPoints. I am a very visual person, so the pictures were helpful. It was also helpful that the other researchers broke down the information into small parts. There were a few confusing moments, but the people at Boston University explained everything well, and I figured it out. The graduate students were really helpful. They helped with activities and hung out and talked with us. They also helped me enter my work hours.

### 2.1.2. Being a Peer Mentor

I had two mentees who I met once a week at their house or school. Other mentors met with their mentees at other places in the community, like libraries or coffee shops. When we first met our mentees, we went over how mentoring would happen. We explained what we would do if something went wrong. We worked on a contract together that we all agreed on. We did lots of activities. During our sessions, I taught them lessons about mental health by doing a worksheet, an activity, or showing a video we made. Some of these activities were a body scan, barriers and support worksheet, and mood logs (Figure 1).

Our goal was to pick two coping strategies with the mentee. We determined what coping strategies our mentee would work on by doing a card sort. The card sort also helped us to find out what their interests were. Examples of coping strategies from my mentees included making origami, a coloring book, or going to the park. After they picked their strategies, we helped them to work on their coping strategy and discussed any worries or concerns they were having. Each week, I also checked in with my mentee by a phone call, text or email about how they were doing.

To help me to be a better mentor, I had a supporter who I talked to weekly about the week's mentoring session. We talked about the script and the activities I would do. Sometimes if the mentoring script was confusing, we would edit it to make it better.

During the peer mentoring group, before we even started meeting with mentees, we talked about what to do in an emergency situation. We had a texting system. We would text a stop sign to our supporter if there was an emergency. We would text a yield sign if there was something I needed to tell my supporter later. I never had to use the stop sign,

but it was there just in case. I thought the texting system was good, and it was helpful that Ariel worked near one of my mentees. Once, when I had a problem, she came running over to help.

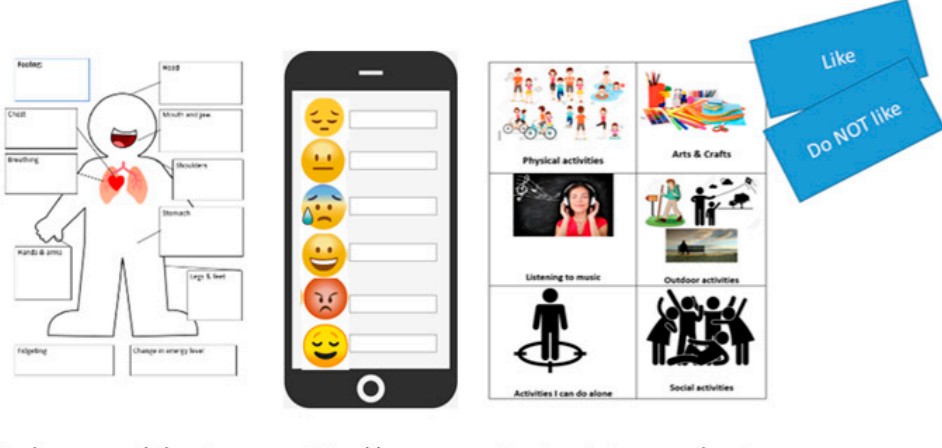

Body scan worksheet          Mood log          Coping strategy card sort

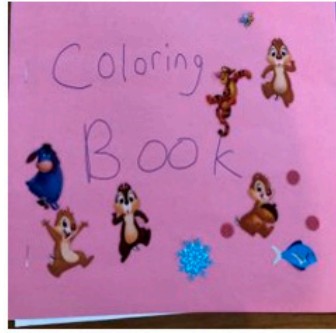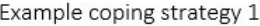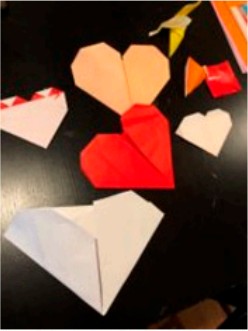

Example coping strategy 1                    Example coping strategy 2

**Figure 1.** Example of peer mentoring activities.

### 2.1.3. Looking at Data to Find Out If Mentoring Helped Our Mentees

After mentoring, we looked at the data. The data included interviews with the mentees and their parents and surveys about mental health symptoms that the mentees and their parents filled out. To analyze the data, we made charts on what the mentees and their family had said. For example, when the research team met at Boston University, we did an activity where we got a bag of quotes showing what people liked and did not like. We worked together with a partner and got help from Ariel and the graduate students to sort quotes from the mentees into categories and glued the quotes on paper (Figure 2). One group worked on positive things about mentoring and another group worked on negative things about mentoring. We presented the charts to each other. Then, we talked about what they liked and did not like as the research team so we could know what to change. When I was looking at the data, I kept thinking back to my actual mentoring sessions. I could remember what my mentee did and how some of the data matched my experiences. I would ask questions if other people had similar experiences during mentoring. This helped me understand the information more.

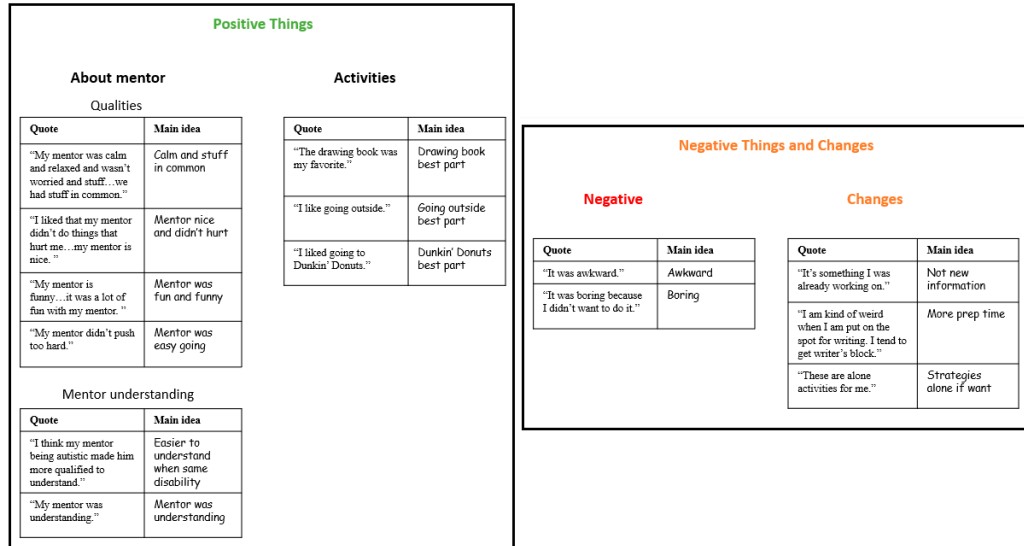

**Figure 2.** Example of the qualitative analysis of feedback about peer mentoring.

We also looked at graphs of the survey data that Ariel made to see if there were changes in their mental health symptoms after mentoring (Figure 3). If I had seen the graphs before I mentored, I would not have understood them as well, because I like to understand things by doing them. When Ariel showed us the graphs, she reminded us of the training we did that related to the different topics in the graphs and how we worked on mental health symptoms in mentoring.

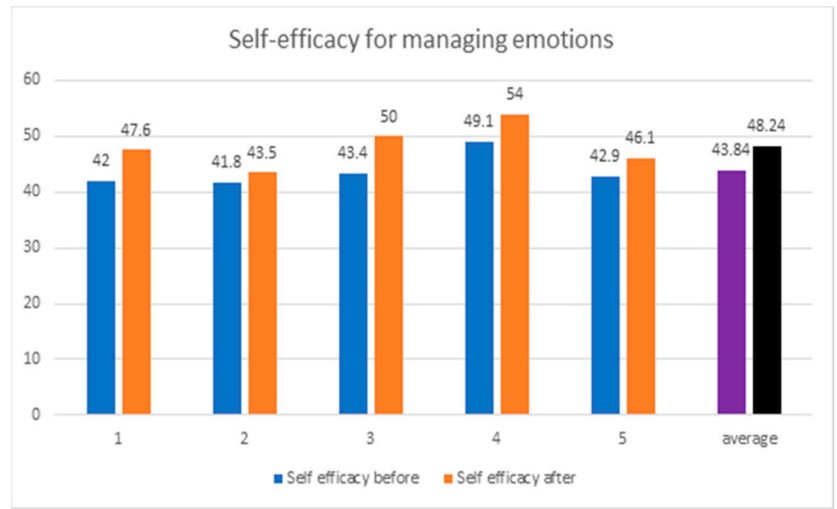

**Figure 3.** Example of an analysis graph.

We talked about making changes to peer mentoring based on the data and what we remembered did and did not work while we were mentors. For example, a couple people said they were getting bored with the barriers and supports worksheet and that it took too long (Table 1). So, we made it into a card sort to make it more fun. We are planning on doing peer mentoring again. Because of the COVID-19 pandemic, we will be doing it virtually and we can have mentors and mentees from outside of Massachusetts.

**Table 1.** Selected quotes about peer mentoring.

| What Mentees Liked about Mentoring | What Mentees Wanted to Change about Mentoring |
|---|---|
| • "[My favorite part was] spending time with Alix and making good conversations . . . she was very calm and relaxed, and she wasn't worried and stuff . . . we had stuff in common . . . she was a very good person. [She made me feel comfortable because] she wasn't yelling, 'cause I don't like yelling . . . she was very understanding".<br>• "[Alix] is funny!"<br>• "My favorite one is going to coffee . . . it was a lot of fun".<br>• "The drawing book is a lot of fun to make".<br>• "I like that [my mentor] is not an ablest person. Because some mental health providers they think that helping people and hurting people are the same thing . . . . And that's an awful thing to have experience with and they can just be terrible and [my mentor] was not like that . . . And I think him also being autistic make him more qualified to understand". | • "Maybe talking more about myself . . . stuff that's on my mind"<br>• "Practicing the coping strategies . . . I think it might be good if we did it more often."<br>• "Spend more time"<br>• "Maybe, a little bit more obscure knowledge . . . instead of trying to teach you about mental health, which you no doubt already know if you are at my age, you already know about it because you grew up around it . . . teaching you how to, take responsibility for yourself, like learn about your rights, as someone who has problems with mental health, things like that". |

### 2.2. What I Learned from Peer Mentoring

In the mentoring project, I learned how to work on a research team. I learned that giving an idea is good. I also learned time management, because if you have a meeting with a mentee you have to be on time. I set alarms on my phone and I made sure I had a ride. Peer mentoring helped me a lot with communication with other people. I opened up to talk with other professionals, like my mentee's school counselor, people at a conference, and someone from another university who asked us to make a podcast (Boston University Mentoring and Research Team 2020).

Mentoring helped me have conversations with people in my personal life. I struggle making friends, because sometimes I do not know how to start a conversation, and this program really helped me with suggesting ideas about how to talk to people. It also helped me to learn cues, like if someone does not want to talk to you, because we talked a lot about how to communicate with the mentees.

*I also learned about other people. For example, boundaries are important. If someone is having a hard time, you want to give them space, and then ask again. Mentoring also helped me learn how to give other people a chance to talk. Sometimes, I get really excited about sharing my ideas. I learned to hold back and give someone else a chance.*

### 2.3. Why It Was Important That I Was Involved

People with disabilities have so many more strengths than they have challenges. Giving someone the opportunity to mentor someone just like them is a good idea because they understand where they are coming from. That person comes from experience not by going to school and getting a degree, but because of their own life experience. I personally feel we need more mentors like me because I have an understanding that cannot be taught in a classroom. I feel I have a connection with my mentees and we probably have had the same struggles. In some ways, we speak the same language. I feel I can truly understand what the mentees are talking about and understand their experiences.

It is important that young people with disabilities helped make the peer mentoring program, instead of just adult researchers without disabilities, because as a person with a disability, I understand their daily struggles. When I was growing up, I needed a lot

of support, academically and emotionally. I really looked up to people who helped me, and I thought it was super important. I really wanted to give back to someone who was struggling just like I was.

## 3. Disclosure Project

### 3.1. What the Study Was about and What I Did

Our research was to learn more about the experiences that young adults with intellectual/developmental disabilities and co-occurring mental health conditions have at work. We wanted to know if they are receiving accommodations they need in the workplace. We also wanted to see what they were having challenges with at work. A big part of this study was learning about disclosure. Disclosure is when you tell someone about your disability. We wanted to know if people disclose their mental health conditions at work, and if they do, how they did it (Interview with Boston University Research Team 2021; Schwartz et al. 2022). In this project we interviewed 12 young adults with intellectual/developmental disabilities and mental health conditions to find out how they disclosed at work and to hear their stories.

To conduct the research, we worked as a team. I was really lucky to work with an amazing team of researchers who helped me every step of the way. They gave advice about what to say during the interviews. We collaborated and shared our ideas as a group. There were three researchers on this team that had an intellectual/developmental disability and mental health conditions. Ariel, who was working at Boston University, was also part of the team. We met every couple of weeks on Zoom. We created the questions that we were going to ask to the participants and also made a survey about accommodations and how mental health impacted young adults at work. We came up with the questions for the survey and response options. We had some ideas for the survey that Ariel said she never would have thought of without us. One of those ideas was to have really specific response options on the survey that would give us more information about accommodations (Figure 4).

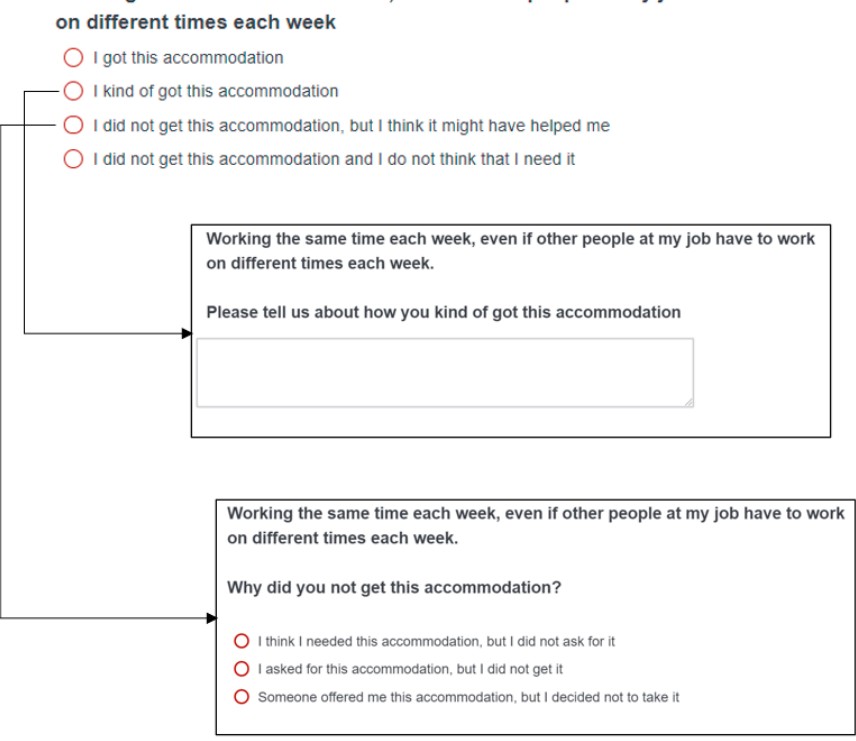

**Figure 4.** Response options developed by the team.

For the interviews, we put all the questions on a Google Slideshow that we shared. When we were interviewing a young adult, there was one researcher and Ariel in the interview. We took turns asking questions and listening to their stories. Ariel would change the color of the question once we were done getting the information we needed. If we needed more information, Ariel privately chatted suggestions for another question to ask, or reminded us about "who", "why", or "how" questions. We had different topics on each slide to make sure that we were asking the correct question. I thought it was really helpful that each slide had a few questions. The Google Slides were on a screen that only me and Ariel could see. We audio recorded all of the interviews and graduate students at Boston University transcribed everything the participants said so we would have all the data. Ariel had also talked to job coaches and transition specialists in focus groups and recorded what they said. We also analyzed that data. We spent a long time gathering the data!

After we interviewed all of the young adults, we analyzed the data by looking at Jamboard to help us to understand it completely. Ariel read the hundreds of pages from the focus groups and interviews and found quotes about different topics. Some examples of the topics are: accommodations, how people disclose, and what happened when people disclosed. She put the quotes on post-it notes in Jamboard. We moved the post-it notes into categories based on the main idea in the quote. Then, we either made more categories or changed the name of the category to help us to understand the data (Figure 5).

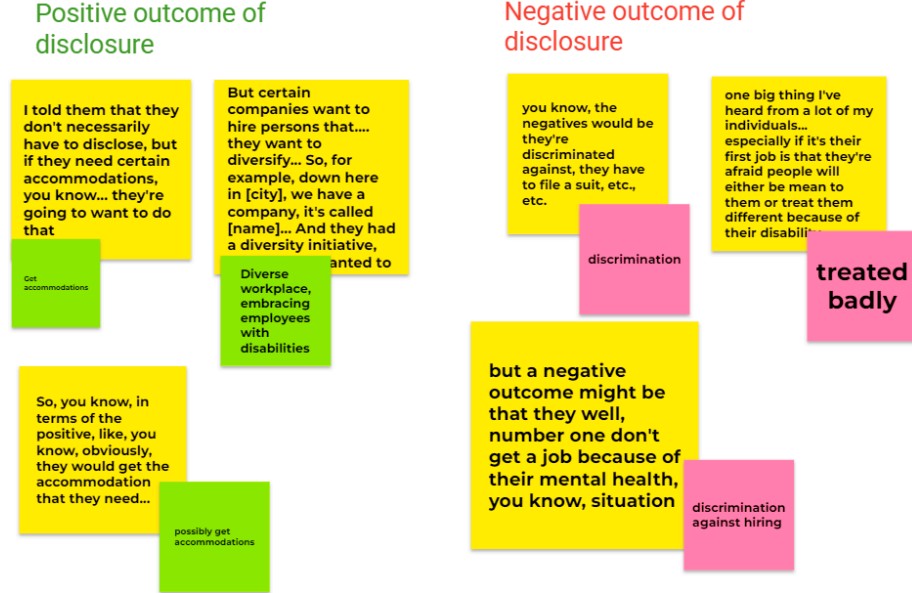

**Figure 5.** Example of the qualitative analysis using Jamboard. Quotes are in yellow; co-researcher's "main idea" labels are in green and pink.

I really liked this project because I enjoy listening to other people's stories and trying to help them as much as I can. It was interesting to me, because I could relate to some young adults and what they shared.

*3.2. What I Learned from the Disclosure Project*

In the disclosure project, I learned a lot about the different jobs young adults had. I learned how they disclosed at work. After doing data analysis I really understood what some people were going through at work and what support they were getting and not getting. I personally think that, if you struggle with something, someone should know about it, so you do not go to a job and have to struggle. This does not mean you have to disclose a disability, but you can ask for help.

### 3.3. Why It Was Important I Was Part of This Project

We are coming from the experience of having a disability. The things that we were researching and talking about are our real-life experiences. I felt like I could relate to some of the concerns the participants were having. During some interviews, after a young adult told us a story about what happened at work, I could say, "that happened to me". That might have made them feel like they are not the only one. As a researcher with a disability, I have been through so many job trainings and have spoken to so many professionals about work that it came naturally to me. This also helped me to come up with interview and survey questions. I felt like I could bring a lot of good ideas that might help other young adults with disabilities. I would be curious if a team of researchers without disabilities found the same things.

## 4. How Research Impacted My Life

Research positively impacted my life. I really liked being a researcher because I have always wanted to help people, and this gave me an opportunity to do that. I really enjoyed listening to the stories the young adults had in the interviews for the disclosure project. It makes me feel happy when other young adults share stories with me because, sometimes, I can relate to them. I also was able to learn about my own strengths and be more confident.

I learned things that helped me in my own life. For example, in peer mentoring, I learned more about communication, and in the disclosure project, I learned about accommodations I could have asked for. I also learned about how to teach other people about coping strategies and mental health. Both studies helped me to learn how to organize information when we made charts or put information in a PowerPoint.

Doing research helped me realize how many strengths I have. For example, I realized that I have strengths in communication, being professional, and listening to what someone is saying. I am very creative. I got to use my creativity by sharing ideas. For example, I gave ideas of what we should put in the video we made about peer mentoring. I also gave ideas about how to improve peer mentoring activities. One idea was making the body scan on the computer instead of on paper.

I also was able to use my communication skills. When I am talking to someone, I always want them to feel that they are heard and that I am acknowledging what they say. For example, when someone tells me a personal story, I say, "thank you so much for sharing that. I know that can be really hard". This was really helpful in research because some of the topics that we were talking about could be sensitive and hard to talk about.

Doing research helped me think of new options for myself. I learned more about helping people instead of being the person who got help. It is very different. Research helped me see that I am a good advocate for young adults with disabilities and that advocating is something I really like to do. When I learned new skills, I felt like I could be a leader. Ariel taught us the skills to lead a mentoring session. To prepare to lead, we looked at PowerPoints and did hands-on activities, such as role plays. These activities helped me understand it right away. I felt like I was the leader when I was mentoring because I was the person teaching.

This job helped me get other jobs. I used to work in food service. Then, I started the mentoring project, and I realized that I was really passionate about working with people with disabilities and mental health conditions. I looked for a job working with people. It took a long time, and that led to some mental health struggles, because I had jobs I did not like. The mentoring program opened a lot of doors for my career. Mentoring gave me the experience I needed. I also asked Ariel to tell my job coach and other people that I could work with kids and teens really well. Now, I am a preschool teacher.

## 5. How Researchers without Disabilities Can Help Researchers with Disabilities Feel Welcome

Researchers without a disability can make researchers with a disability feel welcome by listening to their ideas, supporting their ideas, and including them in group decisions.

Ariel and the other people helped me to feel like I belonged when they listened to my ideas and supported me if I did not understand something. They also used visuals, which helped me a lot. Another way that helped me understand the data was working in small groups. Ariel really broke the data down and explained it to us in a way that we understood. That structure helped me understand it better.

Now that we have worked together for three years, I feel even more comfortable sharing my ideas. I also felt as if I belonged, because other people on the team had disabilities. I could relate to what we were talking about, and I knew how it felt from my own personal experience. I had real life experiences, so I had a lot to say. People can read books about people with disabilities, or watch videos, and do so much research, but the experts are the people that experience disability in their own personal life.

## 6. Conclusions

As a researcher, working on these projects, I felt it was really unique, because I was able to promote and develop the research, and usually people with disabilities do not do things like that. Being a researcher was a very important experience for me. I always knew I wanted to help people, but I did not know what type of people I wanted to help. These experiences helped me learn I wanted to help people with disabilities, just like people helped me when I was younger. Doing research helped me open up about what I think is important and I learned many skills, just like the other co-researchers also learned many skills in their research (e.g., McDonald and Stack 2016; Strnadova et al. 2014; White and Morgan 2012).

As I was working on the research team, I contributed all of my strengths to the group and was an advocate for the things that I felt were important for people like me. Because Ariel designed the research to play to my strengths and understood how important my life experience was, it made it easy to contribute. In this group I felt comfortable sharing my ideas, because Ariel always gave my ideas a try (Milner and Frawley 2018; Schwartz et al. 2020a). After many years of working on the research, I have gained a lot of skills and strengths and learned new information about how to help somebody like me. The longer I do research, the better I become at it and the more I see how I have made a difference in the projects (Kidd et al. 2018). I am very excited to keep working on the projects and making a difference in somebody else's life. It is one of my passions.

**Author Contributions:** Conceptualization, A.H. and A.E.S.; writing—original draft preparation, review, and editing, A.H. and A.E.S.; funding acquisition, A.E.S. While research described was completed with A.E.S. was at Boston University, conceptualization and writing of this manuscript occurred while she held a position at the MGH Institute of Health Professions. All authors have read and agreed to the published version of the manuscript.

**Funding:** While the team had no funding to support the writing of this manuscript, the studies Alix described were funded by grants to Ariel Schwartz. Peer mentoring study: Deborah Munroe Noonan Memorial Research Foundation & American Academy for Cerebral Palsy and Developmental Medicine Research Grant. Disclosure study: Analyzing Relationships between Disability, Rehabilitation and Work Small Grant Program; Society Security Administration and Policy Research, Inc.

**Institutional Review Board Statement:** The studies described were conducted in accordance with the Declaration of Helsinki, and approved by the Institutional Review Board at Boston University (Protocol 4903E, approved 10 August 2018 and Protocol 5658E, approved 21 August 2020).

**Informed Consent Statement:** Informed consent was obtained from all participants involved in the described studies.

**Data Availability Statement:** Not applicable.

**Acknowledgments:** Thank you to everyone on our research team for being great teammates. Thank you to all the participants who were involved in helping to answer our research questions. In addition, we would like to thank the graduate students from Boston University who helped us with our research.

**Conflicts of Interest:** The authors declare no conflict of interest.

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
