# Peer review of "How Being a Researcher Impacted My Life"

_socsci, doi:10.3390/socsci11030127_

Round 1

Reviewer 1 Report

It is a huge honor to be able to review this article. Thank you for the invitation. The article touched me very much because it touches on a very important topic. It shows how important the feeling of belonging is to a group of disabled people and how important scientific work is for them. The authors present in a beautiful, legible way how they participated in the projects and how much they learned. I am very impressed with the creativity in the activities shown in Figure 1. I suggest the authors of the article include in their work an analysis of whether such projects have already been implemented by people with disabilities, and if so, when and what they were related to. Thank you for being able to be part of this beautiful scientific work.

Author Response

Dear Reviewer 1,

Thank you for your feedback and comments. We appreciate your enthusiasm and support.

In the introduction, we added some information about previous papers written by co-researchers, including citations. In particular, we reference a review paper that addressed the topic of co-researcher voice. We did not extensively review the information shared in these other papers, because we thought it may take up too much space, and add too much information that was not related to Alix’s main reflections. We can add more information if you think more is needed.

Thank you again for your very kind words.

Reviewer 2 Report

This article is a reflective writing of personal experiences gained in research projects. The paper does not have a data (several tables and examples of the discussed project are presented). I simply do not see this as a research paper. It could be considered for publication as a “discussion” or “reflection” article but not as an original research. I am happy that the author has gained positive experiences working in research profession but these reflective examples do not relate to any other than the author oneself. Unfortunately, I cannot recommend publishing this in an international journal.

Author Response

We have not responded to this reviewer, as the editor seemed to indicate it was not necessary.

Reviewer 3 Report

Dear Authors,

Thank you so much for giving me the chance to review your interesting manuscript. Interesting and also adding to our body of knowledge, because very little is written from the perspective of a researcher with intellectual disabilities (ID), and even less by a researcher with ID themselves, being an author. Congratulations to the author team for having accomplished this. It is a truly authentical article that not only gives much insight in the process of being a researcher with ID, but is also written in a very accessible way. I like the clear process that you have followed to make this happen. Next to this, I liked the figures a lot, they were very helpful in giving me a picture of the steps you have taken. (I am a visual person myself). The lessons learned from being a researcher for your own life (page 9) are so important! Every researcher learns something for or about their own life from projects, especially the ones that you are passionate about. And we think so often in academia we are not 'objective' in reflecting on the impact of our research projects on our own lives. Paragraph 5 is (therefore) also very important and helpful. You have inspired me to encourage people with ID/DD to publish in their own way. 

I have a few suggestions to strengthen your article:

  1. the study on Disclosure is more reflecting the process of being a researcher than the study on Peer Mentoring. I was wondering why, and I think it has to do with the contamination in this written text of a) researching Peer Mentoring as a programme and b) being a mentor yourself. What I understand from it is that you have to do a) to be able to do b). My suggestion is that you clarify this in the text and see what you need from part a) as illustration for part b).
  2. Maybe it is helpful to reduce the 'being a mentor part' and explain why this was an important step in the research part of the project (that is your focus in this article). Between this paragraph (2.2.2. on page 2) and the next paragraph (2.2.3 on page 3) I was missing information on how the charts were made. Were they made by the two of you, or by the whole team? What was the process that you followed? 
  3. In paragraph 2.2.3, p. 3 line 102: whom were part of this group? mentees and families, the research group, other persons?
  4. on p. 7 you describe the GoogleSlideShow as a way that worked for you in putting interview questions together. I was wondering if that worked for everyone that was being interviewed? For instance, I have worked with a person who is visually disabled, and they is not happy with visuals. 
  5. The conclusions are written in a different (more 'objective') style, that distracted me from the engaging style of the body of the article. I recommend to rewrite this paragraph in a similar way to be consistent all the way.

Author Response

Dear Reviewer 3,

Thank you for your feedback and comments. We appreciate your enthusiasm and support of our work!

Below, we respond to each comment

  1. The study on Disclosure is more reflecting the process of being a researcher than the study on Peer Mentoring. I was wondering why, and I think it has to do with the contamination in this written text of a) researching Peer Mentoring as a programme and b) being a mentor yourself. What I understand from it is that you have to do a) to be able to do b). My suggestion is that you clarify this in the text and see what you need from part a) as illustration for part b).
  2. Maybe it is helpful to reduce the 'being a mentor part' and explain why this was an important step in the research part of the project (that is your focus in this article). Between this paragraph (2.2.2. on page 2) and the next paragraph (2.2.3 on page 3) I was missing information on how the charts were made. Were they made by the two of you, or by the whole team? What was the process that you followed? 

We responded to point 1 and point 2 together, because we thought that the comments went together. We think that the reviewer wanted a better explanation of how being a mentor helped Alix be a better researcher. We added more information on p.4-5 to explain how Alix’s experiences mentoring helped her interpret the data and make suggestions for changes to mentoring.

We also explained who made the graphs and more details about the charts.

3. In paragraph 2.2.3, p. 3 line 102: whom were part of this group? mentees and families, the research group, other persons?

This is a good question. Thanks for helping us see that our writing was not as clear as it could have been. We added that this group was our research team.

4. On p. 7 you describe the GoogleSlideShow as a way that worked for you in putting interview questions together. I was wondering if that worked for everyone that was being interviewed? For instance, I have worked with a person who is visually disabled, and they is not happy with visuals. 

We explained that the slides were only shared between the academic researcher and Alix, and not with the participants. We hope this clarification helps the reviewer. We would have had to figure something else out if Alix had a visual impairment. However, we felt that talking a lot about this would be somewhat off topic from Alix’s experiences. We both agreed that this was a very good point!

5. The conclusions are written in a different (more 'objective') style, that distracted me from the engaging style of the body of the article. I recommend to rewrite this paragraph in a similar way to be consistent all the way.

Thank you for this excellent suggestion. We were nervous about not sounding “academic enough,” so the academic-researcher wrote this paragraph. We agree it much better in Alix’s voice.